# Hepatitis E seroprevalence in a German cohort of patients with inflammatory bowel diseases

**Peter Hoffmann**[1]*, **Rouven Behnisch**[2], **Julia Gsenger**[3], **Paul Schnitzler**[3], **Annika Gauss**[1]

**1** Department of Gastroenterology and Hepatology, Heidelberg University Hospital, Heidelberg, Germany, **2** Department of Medical Biometry, Institute of Medical Biometry and Informatics, Heidelberg University Hospital, Heidelberg, Germany, **3** Center for Infectious Diseases, Virology, Heidelberg University Hospital, Heidelberg, Germany

* peter.hoffmann@med.uni-heidelberg.de

**Data Availability Statement:** All relevant data are within the manuscript and its Supporting Information files.

## Abstract

### Background and aims

The incidence of hepatitis E virus (HEV)-infections in industrialized nations has been increasing over the past years. Patients suffering from inflammatory bowel diseases (IBD) may be more prone to transmission. Data on HEV seroprevalence in IBD patients is scarce and has not been reported in German IBD patients. The German Health Examination Survey for Adults 2008–2011, which included 4.422 samples, found a HEV seroprevalence of 16.8%, increasing with age. The aim of the present study was to determine the seroprevalence of anti-HEV IgG in a German cohort of IBD patients, and to explore which parameters have an impact on HEV seroprevalence.

### Material and methods

This is an uncontrolled, cross-sectional, retrospective monocentric study. Among the patients visiting the IBD outpatient clinic between 25 January, 2019 and 24 September, 2019, 328 patients with Crohn's disease (CD) and 150 patients with ulcerative colitis (UC) were included in the study. IgG antibodies against HEV were determined by enzyme-linked immunosorbent assay. Positive antibody titers were verified using immunoblot analysis. Medical records were reviewed for demographic and clinical parameters to identify potential risk factors for HEV infection.

### Results

The prevalence of anti-HEV IgG antibodies was 17.4% in CD patients and 24.7% in UC patients. No patient with positive HEV PCR was detected. Greater age of CD und UC patients and longer duration of anti-interleukin 12/23 treatment in CD patients were associated with higher anti-HEV IgG antibody rates.

**Funding:** The author(s) received no specific funding for this work.

**Competing interests:** Annika Gauss received travel fees from Abbvie and Janssen, speaker's fees from Janssen, MSD, Tillotts, and Takeda, and consultation fees from Janssen and AMGEN; Peter Hoffmann received travel fees from Abbvie and Janssen and speaker's fees from Janssen; which might lead to a conflict of interest. This does not alter our adherence to PLOS ONE policies on sharing data and materials. All other authors have no conflicts of interest to declare.

## Conclusions

In summary, we conclude that patients with UC have a higher anti-HEV IgG antibody prevalence than the general population in Germany, and that immunosuppressive therapy may carry no higher risk for HEV infection.

## Introduction

The number of hepatitis E virus (HEV) infections in industrialized nations has been increasing over the past years. The anti-HEV IgG seroprevalence is 6.8% in German blood donors [1] and 16.8% in the German population [2], while it is 16% in English blood donors [3]. Most HEV infections are asymptomatic, but they may become chronic in patients under immunosuppressive therapy. Acute symptomatic HEV infections have previously been reported in Germany [4–6]. In a multicentric European cohort of 21 internal medicine patients with various disturbances of their immune system, seven (33%) developed chronic HEV infection [7].

A relatively high rate of HEV viremia of 0.12% was found in German blood donors [8] and of 0.076% in Dutch blood donors [9]. Thus, obligatory HEV-testing of blood donors has been introduced in Germany in January 2020. In a cohort of Dutch blood donors, eating raw meat and sausages were sources of HEV infection [10]. Patients with inflammatory bowel diseases (IBD) may be at higher risk of HEV infection due to a disturbed intestinal barrier function and immunosuppressive medications, such as glucocorticosteroids, thiopurines, and anti-tumor necrosis factor (TNF) agents or other biologicals. Data on HEV seroprevalence in IBD patients is scarce, and not available in Germany. Varying dietary habits in different countries may explain regionally different anti-HEV IgG seroprevalences. The only study we found on anti-HEV IgG seroprevalences in IBD patients originates from Spain. Its authors describe a low anti-HEV IgG seroprevalence of 1.14%, corresponding to the HEV seroprevalence found in the general population in Spain [11].

HEV infection can be caused by four different genotypes auf HEV. Genotypes 3 and 4 are responsible for 90% of all cases in industrialized countries [12]. As hepatitis E is a zoonosis in industrialized countries, transmission may take place via the consumption of raw meat or close contact to certain animals [13]. Lifestyle behaviors, such as poor dietary and household hygiene, may also play a role for transmission of HEV [14].

In patients suffering from liver cirrhosis and in pregnant women in Africa and Asia, the virus has been described to cause liver failure in some cases [15]. In patients under immunosuppressive therapy like patients after organ transplantation, high rates of chronic HEV infection were found [16, 17].

The aim of this study was to determine the seroprevalence of anti-HEV IgG in IBD patients, and to investigate whether factors such as disease duration and quality or duration of immunosuppressive therapy have an impact on HEV seroprevalence in these patients.

## Material and methods

This is an uncontrolled, cross-sectional, retrospective monocentric study including outpatients suffering from IBD at a German university hospital. The study protocol was approved by the local Ethics Committee (Ethikkommission Heidelberg, Alte Glockengießerei 11/1, 69115 Heidelberg, protocol number: S-324/2019).

Inclusion criteria were as follows: 1. having consented to participate in the local biobank for IBD patients; 2. having visited the IBD outpatient clinic of Heidelberg University Hospital between January 25, 2019 and September 24, 2019; 3. suffering from Crohn's disease (CD) or ulcerative colitis (UC). The diagnosis was made based on ECCO criteria [18].

Exclusion criteria were: 1. age < 18 years, and 2. denial to take part in the IBD biobank sampling.

Serum samples from the included patients were used for determination of anti-HEV IgG antibodies. Antibodies against HEV were determined by ELISA (recomWell HEV IgG, recom-Well HEV IgM, Mikrogen, Neuried, Germany). Positive antibody titers were further analyzed by immunoblot (recomLine HEV IgG/IgM, Mikrogen, Neuried, Germany) in order to determine the specificity of the anti-HEV antibodies detected by ELISA.

In serum samples from patients with anti-HEV IgG antibodies PCR for HEV RNA was performed.

For PCR analysis, RNA was extracted from serum using the QIAamp viral RNA mini kit (Qiagen, Hilden, Germany) according to the manufacturer's protocol. Amplification and detection of viral RNA was performed with the RealStar HEV real-time PCR kit (altona Diagnostics, Hamburg, Germany) on a LightCycler 480 instrument II (Roche, Mannheim, Germany).

Medical records were reviewed for demographic and clinical parameters (gender, age at HEV antibody determination, history of bowel resections, disease phenotypes, disease duration) and therapy with immunomodulators (azathioprine, 6-mercaptopurine, methotrexate, tofacitinib), or biologicals (anti-TNFα, anti-interleukin 12/23, anti-integrin). We also investigated the occurrence of increased plasma liver enzyme concentrations within the total time span during which the patients were visiting the IBD outpatient clinic, using a cut-off time span of 10 years prior to anti-HEV IgG antibody measurement. Transaminases > 2-fold of upper limit were defined as increased. The Montreal classification for CD [19] was used to categorize disease phenotypes.

Descriptive statistics were calculated as percentages for discrete variables and presented as medians with ranges. To identify potential risk factors for HEV infection, the Mann-Whitney test was used for ordinal and continuous variables, and Chi-squared tests for categorical variables. For multivariable analysis, factors that were univariately associated with the outcome with a P-value of 0.05 or less were included in a logistic regression model with variable selection.

Due to the exploratory nature of the study, P-values are to be interpreted in a descriptive manner. Thus, no adjustment for multiple testing was performed. P-values < 0.05 were regarded as statistically significant. All statistical analyses were performed using IBM SPSS Statistics 24 (Chicago, IL, USA).

## Results

### Crohn's disease

In total, 430 CD patients have been included in the IBD biobank sampling. Among the CD patients included in the IBD biobank, data on HEV prevalence was not available for 95 patients: in 40 cases, this was due the fact that taking serum samples for anti-HEV IgG determination was missed, and 55 patients did not visit the IBD outpatient clinic between January and September 2019.

In total, 335 serum samples from CD patients were analyzed for the presence of anti-HEV IgG. Seven CD patients had to be excluded due to insufficient documentation of the history of medical therapy (Fig 1).

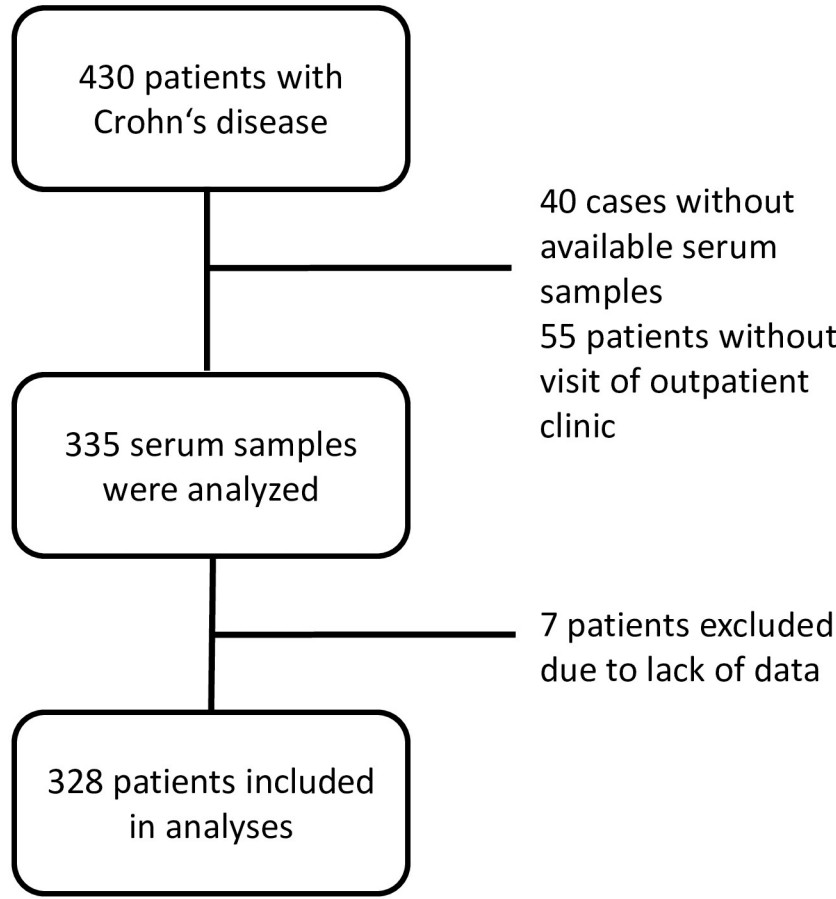

**Fig 1. Flow diagram of inclusion and exclusion in Crohn's disease patients.**

Three hundred twenty-eight CD patients (148 males) with a median age of 44 years were included in the study. In total, 88.1% of these patients had a history of immunomodulator or biological therapy, and 65.5% had undergone bowel resection(s). The median disease duration until determination of HEV serology was 13 years. Details are shown in Table 1.

Anti-HEV IgG antibodies were found in the serum samples of 17.4% CD patients. In these patients, no HEV RNA was detected. Age, disease duration and duration of anti-interleukin 12/23 treatment at anti-HEV IgG determination were significantly higher in anti-HEV IgG positive patients as compared to anti-HEV IgG negative patients in a univariate analysis (Table 2).

There was no higher prevalence of elevated plasma transaminase concentrations in patients with anti-HEV IgG antibodies versus those without proof of antibodies during the time the patients were treated at our outpatient clinic (21.1% vs 22.5%, P = 0.81). A multivariable logistic regression model including the parameters reaching P<0.05 in the univariate analysis identified age at HEV serology (P = 0.002) and duration of anti-interleukin 12/23 treatment (P = 0.035) as risk factors for HEV infection in CD patients (Table 3).

### Ulcerative colitis

In total, 212 UC patients have been included in the IBD biobank sampling. For 59 among the UC patients included in the IBD biobank, HEV serology was not available for the following

**Table 1. Characteristics of all included patients.**

| | UC | CD |
|---|---|---|
| **Variable** | **n = 150** | **n = 328** |
| Male, n (%) | 80 (53.3) | 148 (45.1) |
| Age at diagnosis of IBD (years), median (range) | 29 (11–68) | 24.5 (3–75) |
| Age at HEV serology (years), median (range) | 45 (18–86) | 44 (18–78) |
| Disease duration at HEV serology (years), median (range) | 10 (0–49) | 13 (0–50) |
| History of bowel resection(s), n (%) | 26 (17.3) | 215 (65.5) |
| History of anti-TNFα treatment, n (%) | 72 (48.0) | 238 (72.6) |
| History of anti-interleukin 12/23 treatment, n (%) | 1 (0.7) | 73 (22.3) |
| History of anti-integrin treatment, n (%) | 52 (34.7) | 58 (17.7) |
| History of any biological treatment, n (%) | 95 (63.3) | 249 (75.9) |
| History of immunomodulator treatment, n (%) | 77 (51.3) | 211 (64.3) |
| History of JAK-inhibitor treatment, n (%) | 13 (8.7) | 0 (0) |
| History of any immunomodulator or biological treatment, n (%) | 113 (75.3) | 289 (88.1) |
| **Montreal classification**: | | |
| Age, n (A1:A2:A3) | 9:101:40 | 30:251:47 |
| Location, n (E1:E2:E3), n = 139 | 11:45:83 | |
| Location, n (L1:L2:L3:L4), n = 326 | | 109:42:175:32 |
| Behavior, n (B1:B2:B3), n = 326 | | 114:80:132 |
| Perianal, n = 326 (%) | | 93 (28.4) |

CD: Crohn's disease; HEV: hepatitis E virus; IBD: inflammatory bowel disease; JAK: Janus kinase; TNFα: Tumor necrosis factor alpha; UC: ulcerative colitis.

reasons: in 18 patients, taking a serum sample within the indicated time frame was missed; 41 UC patients did not visit the IBD outpatient clinic between January and September 2019.

One hundred fifty-three serum samples from UC patients were analyzed for the presence of anti-HEV IgG antibodies. Three UC patients had to be excluded due to poor documentation of the history of medical therapy (Fig 2).

One hundred fifty UC patients (80 males) with a median age of 45 years were included in the statistical analyses. In total, 75.3% of these patients had a history of immunomodulator or biological therapy, 17.3% had a history of bowel resection(s). The median disease duration until HEV serology determination was 10 years. Details are presented in Table 1.

Anti-HEV IgG antibodies were detected in the serum samples from 24.7% of the included UC patients. In these patients, no HEV RNA was detected. Age, age at diagnosis and disease duration were significantly higher in anti-HEV IgG positive patients than in anti-HEV IgG negative patients in the univariate analysis, while there was no difference concerning quality and duration of immunosuppressive treatment (Table 4).

There was no higher rate of elevated plasma transaminase concentrations in patients with anti-HEV IgG antibodies versus those without proof of antibodies during the time that the patients were visiting our outpatient clinic (24.3% vs 17.0%, P = 0.32).

A multivariable logistic regression model including the parameters with P<0.05 in the univariate analysis identified age at HEV serology (P = 0.001) as a risk factor for HEV infection in CD patients (Table 5).

**Table 2. Comparison of characteristics between the subgroups of patients with positive anti-HEV IgG serology and negative anti-HEV IgG serology in Crohn's disease.**

| Variable | | anti-HEV IgG negative | anti-HEV IgG positive | P |
|---|---|---|---|---|
| | | n = 271 | n = 57 | |
| Male, n (%) | | 120 (44.3) | 28 (49.1) | 0.504[1] |
| Age at diagnosis of CD (years), median (range) | | 24 (8–70) | 28 (3–75) | 0.140[2] |
| Age at HEV serology (years), median (range) | | 42 (18–78) | 53 (23–78) | 0.000[2] |
| Disease duration at HEV serology (years), median (range) | | 13 (0–49) | 21 (1–50) | 0.012[2] |
| History of anti-TNFα treatment, n (%) | | 194 (71.6) | 44 (77.2) | 0.389[1] |
| History of anti-interleukin 12/23 treatment, n (%) | | 55 (20.3) | 18 (31.6) | 0.063[1] |
| History of anti-integrin treatment, n (%) | | 46 (17.0) | 12 (21.1) | 0.463[1] |
| History of any biological treatment, n (%) | | 204 (75.3) | 45 (78.9) | 0.556[1] |
| History of immunomodulator treatment, n (%) | | 174 (64.2) | 37 (64.9) | 0.919[1] |
| History of any immunomodulator or biological treatment, n (%) | | 241 (88.9) | 48 (84.2) | 0.317[1] |
| **Montreal classification of CD:** | | | | |
| | Age | | | 0.449[1] |
| | A1 | 26 (9.6) | 4 (7.0) | |
| | A2 | 209 (77.1) | 42 (73.7) | |
| | A3 | 36 (13.3) | 11 (19.3) | |
| | Location (n = 326) | | | 0.839[1] |
| | L1 | 92 (34.1) | 17 (30.4) | |
| | L2 | 35 (13.0) | 7 (12.5) | |
| | L3 | 143 (53.0) | 32 (57.1) | |
| | L4 | 29 (10.7) | 3 (5.3) | 0.209[1] |
| | Behavior (n = 326) | | | 0.928[1] |
| | B1 | 94 (34.9) | 20 (35.1) | |
| | B2 | 65 (24.2) | 15 (26.3) | |
| | B3 | 110 (40.9) | 22 (38.6) | |
| | Perianal disease (n = 326) | 81 (30.1) | 12 (21.1) | 0.169[1] |
| anti-TNFα treatment (days), median (range) | | 629 (0–7020) | 518 (0–4127) | 0.928[2] |
| anti-interleukin 12/23 treatment (days), median (range) | | 0 (0–852) | 0 (0–1931) | 0.040[2] |
| anti-integrin treatment (days), median (range) | | 0 (0–1299) | 0 (0–1615) | 0.373[2] |
| any biological treatment (days), median (range) | | 913 (0–7020) | 978 (0–4127) | 0.557[2] |
| immunomodulator treatment (days), median (range) | | 304 (0–10227) | 122 (0–6056) | 0.592[2] |
| any immunomodulator or biological therapy (days), median (range) | | 1946 (0–13334) | 1979 (0–7456) | 0.761[2] |
| History of bowel resection(s), n (%) | | 173 (63.8) | 42 (73.7) | 0.155[1] |

IgG: immunoglobulin G; HEV: hepatitis E virus; CD: Crohn's disease; SD: standard deviation; TNFα: Tumor necrosis factor alpha;

[1] Chi-square test;

[2] Mann-Whitney-test.

## Discussion

IBD patients represent an immunosuppressed subpopulation at higher risk for infections due to a damaged intestinal mucosa and often also due to immunosuppressive therapy. The key finding of this study is that the prevalence of anti-HEV IgG antibodies was 17.4% in CD patients and 24.7% in UC patients while there was no detection of HEV RNA in any anti-HEV

**Table 3. Results of multivariable logistic regression model to identify risk factors for HEV infection in Crohn's disease patients.**

| Parameter | Odds Ratio (OR) | 95% CI | P-value |
|---|---|---|---|
| Age at HEV serology | 1.040 | [1.015–1.066] | 0.002 |
| Disease duration at HEV serology | 1.012 | [0.985–1.040] | 0.380 |
| Duration of anti-interleukin 12/23 treatment | 1.001 | [1.000–1.003] | 0.035 |

HEV: hepatitis E virus.

IgG antibody positive patient. The prevalence of anti-HEV IgG antibodies tended to be higher in UC patients as compared to CD patients.

In comparison, the German Health Examination Survey for Adults 2008–2011, which included 4.422 samples, found a HEV seroprevalence of 16.8%, increasing with age [2]. The median age in Germany is 46.5 years [20], and the average age in 2011 was 43.9 years [21]. In our cohort, the median age was 44 years in CD patients and 45 years in UC patients. Thus, the data of both cohorts are comparable. In accordance with the results from the above cited study, greater age was associated with higher HEV seroprevalence in our study.

A large systematic meta-analysis of anti-HEV IgG antibody seroprevalence including 29 German studies revealed an anti-HEV IgG seroprevalence of 6–10% [22]. In comparison to these studies, we may assume that HEV seroprevalence in an IBD cohort is higher than in the general population.

However, the choice of the HEV serology test might influence the results and may also explain different results of anti-HEV IgG antibody seroprevalences in different publications [23, 24].

In our study, the seroprevalence of anti-HEV IgG tended to be higher in UC patients than in CD patients. To explain this slight difference, it may be speculated that a defect of the

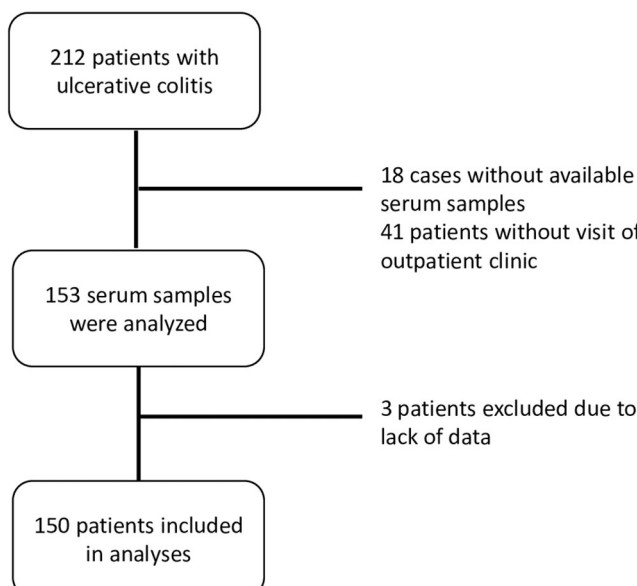

**Fig 2. Flow diagram of inclusion and exclusion in ulcerative colitis patients.**

**Table 4. Comparison of characteristics between the subgroups of patients with positive anti-HEV IgG serology and negative anti-HEV IgG serology in ulcerative colitis.**

| Variable | | Anti-HEV IgG negative | Anti-HEV IgG positive | P |
|---|---|---|---|---|
| | | n = 113 | n = 37 | |
| Male, n (%) | | 60 (53.1) | 20 (54.1) | 0.919[1] |
| Age at diagnosis of UC (years), median (range) | | 27 (11–68) | 40 (14–63) | 0.003[2] |
| Age at HEV serology (years), median (range) | | 40 (18–86) | 55 (21–72) | <0.001[2] |
| Disease duration at HEV serology (years), median (range) | | 10 (0–49) | 13 (1–45) | 0.038[2] |
| History of bowel resection(s), n (%) | | 21 (18.6) | 5 (13.5) | 0.479[1] |
| History of anti-TNFα treatment, n (%) | | 57 (50.4) | 15 (40.5) | 0.295[1] |
| History of anti-interleukin 12/23 treatment, n (%) | | 1 (0.9) | 0 (0) | 0.566[1] |
| History of anti-integrin treatment, n (%) | | 40 (35.4) | 12 (32.4) | 0.742[1] |
| History of any biological treatment, n (%) | | 75 (66.4) | 20 (54.1) | 0.177[1] |
| History of immunomodulator treatment, n (%) | | 58 (51.3) | 19 (51.4) | 0.998[1] |
| History of JAK-inhibitor treatment, n (%) | | 12 (10.6) | 1 (2.7) | 0.137[1] |
| History of any immunomodulator or biological treatment, n (%) | | 88 (77.9) | 25 (67.6) | 0.207[1] |
| **Montreal classification of UC:** | | | | |
| | Age | | | 0.002[1] |
| | A1 | 8 (7.1) | 1 (2.7) | |
| | A2 | 83 (73.5) | 18 (48.6) | |
| | A3 | 22 (19.5) | 18 (48.6) | |
| | Location, n = 139 | | | 0.835[1] |
| | E1 | 8 (7.8) | 3 (8.3) | |
| | E2 | 32 (31.1) | 13 (36.1) | |
| | E3 | 63 (61.2) | 20 (55.6) | |
| anti-TNFα treatment (days), median (range) | | 30 (0–3586) | 0 (0–3953) | 0.416[2] |
| anti-interleukin 12/23 treatment (days), median (range) | | 0 (0–118) | 0 (0) | 0.578[2] |
| anti-integrin treatment (days), median (range) | | 0 (0–1690) | 0 (0–2741) | 0.874[2] |
| any biological treatment (days), median (range) | | 270 (0–3586) | 257 (0–3953) | 0.861[2] |
| immunomodulator treatment (days), median (range) | | 30 (0–7121) | 33 (0–4049) | 0.832[2] |
| JAK-inhibitor treatment (days), median (range) | | 0 (0–355) | 0 (0–159) | 0.153[2] |
| any immunomodulator or biological therapy (days), median (range) | | 610 (0–7138) | 800 (0–6494) | 0.728[2] |

IgG: immunoglobulin G; HEV: hepatitis E virus; JAK: Janus kinase; TNFα: Tumor necrosis factor alpha; UC: ulcerative colitis;

[1] Chi-squared test;

[2] Mann-Whitney-test.

**Table 5. Results of multivariable logistic regression model to identify risk factors for HEV infection in ulcerative colitis patients.**

| Parameter | Odds Ratio (OR) | 95% CI | P-value |
|---|---|---|---|
| Age at HEV serology | 1.051 | [1.021–1.082] | 0.001 |
| Disease duration at HEV serology | 0.999 | [0.960–1.039] | 0.965 |

HEV: hepatitis E virus.

intestinal mucosa might result in a greater susceptibility to HEV transmission and thus cause higher infection rates.

Interestingly, the duration of anti-interleukin 12/23 treatment in CD patients was weakly associated with a higher anti-HEV IgG seroprevalence. So far, nothing is known about a potential link between anti-interleukin 12/23 treatment and HEV infection. Anti-interleukin 12/23 treatment was the most recently approved medical therapy for CD, and it was not used as first-line immunosuppression in our patients. Therefore, we suggest that this subgroup in our study represents a subpopulation of CD patients with most severe disease and a higher probability of having received blood transfusions in the past blood products. This may be the reason for a higher anti-HEV IgG seroprevalence in this specific subgroup of patients in our study.

Retrospectively, we found no higher rate of increased plasma transaminase concentrations in anti-HEV IgG positive patients as compared to anti-HEV IgG negative patients during the time span prior to anti-HEV IgG determination.

Limitations of this study are its retrospective study design, and the fact that we were not able to include serum HEV RNA and IgM levels in our study.

Another drawback possibly resulting in a bias is that we were not aware of the food consumption habits of the included patients. Raw meat and sausages can still be a source of HEV, and vegetarians are less often HEV seropositive [25].

In summary, we conclude that patients with UC have a higher anti-HEV IgG antibody prevalence than the general population, but that immunosuppressive therapy in IBD patients seems to carry no higher risk for HEV infection.

## Supporting information

**S1 File. HEV DNA.**
(XLSX)

## Author Contributions

**Conceptualization:** Peter Hoffmann.

**Data curation:** Peter Hoffmann, Julia Gsenger.

**Formal analysis:** Rouven Behnisch.

**Project administration:** Annika Gauss.

**Supervision:** Annika Gauss.

**Writing – original draft:** Peter Hoffmann, Annika Gauss.

**Writing – review & editing:** Paul Schnitzler, Annika Gauss.

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
