## [Decision Letter · Decision Letter 0]

31 Jul 2020

PONE-D-20-21089

Hepatitis E seroprevalence in a German cohort of patients with Inflammatory Bowel Diseases

PLOS ONE

Dear Dr. Hoffmann,

Thank you for submitting your manuscript to PLOS ONE. After careful consideration, we feel that it has merit but does not fully meet PLOS ONE’s publication criteria as it currently stands. Therefore, we invite you to submit a revised version of the manuscript that addresses the points raised during the review process.

Your manuscript was reviewed by 2 experts in the field. Both identified many important problems in your submission which require your careful attention. Please review the attached comments and provide point-by-point responses. 

We look forward to receiving your revised manuscript.

Kind regards,

Yury E Khudyakov, PhD

Academic Editor

PLOS ONE

Journal Requirements:

2. Please provide the full name of the ethics committee which approved this study in the methods section of your manuscript.

3.Thank you for stating the following in the Competing Interests section:

[Annika Gauss received travel fees from Abbvie and Janssen, speaker’s fees from Janssen, MSD, Tillotts, and Takeda, and consultation fees from Janssen and AMGEN; Peter Hoffmann received travel fees from Abbvie and Janssen and speaker’s fees from Janssen; which might lead to a conflict of interest. All other authors have no conflicts of interest to declare.].

Reviewers' comments:

Reviewer's Responses to Questions

**Comments to the Author**

1. Is the manuscript technically sound, and do the data support the conclusions?

Reviewer #1: Partly

Reviewer #2: Yes

2. Has the statistical analysis been performed appropriately and rigorously? 

Reviewer #1: No

Reviewer #2: Yes

3. Have the authors made all data underlying the findings in their manuscript fully available?

Reviewer #1: Yes

Reviewer #2: Yes

4. Is the manuscript presented in an intelligible fashion and written in standard English?

Reviewer #1: Yes

Reviewer #2: Yes

5. Review Comments to the Author

Reviewer #1: This study investigates the anti HEV seroprevalence in patients with infallamtory bowel disease. The anti HEV seroprevalence in patients with M. Crohn was lower in comparison to patients with Colitis ulcerosa. I calculated the p-value for this finding (Chi Square) and this gave 0.05, borderline significance....

A comparison to previously published cohorts of the German general population or healthy blood donors is difficult. The Mikrogen tests have been strongly improved in the last 5 years and currently these assays have a high sensitivity and specifity..... but 10 years ago these serological tests had some problems..... thus you cannot compare all published data easily....

Furthermore there are geographical variations within Germany....

if you want to compare your data with healthy people, you need your own cohort and not previously published data from various areas, using various assays....

The seroprevalence was associated with anti-interleukin-treatment..... Patients getting this treatment are usually suffering from a more severe course, thus it is more likely they get more frequently blood products or other medical interventions harboring the risk of HEV transmission.

General testing of blood donations for HEV has been started in Germany in January 2020.

If you want to detect chronic HEV infection you need to test by PCR.... I suggest to do this to improve the project....

Reviewer #2: This is a well written manuscript which describes a straightforward study on the possible relation between HEV seroprevalence and inflammatory bowel disease (IBD) in two groups of patients, those with Crohn’s disease (CD) and those with ulcerative colitis (UC).

The data are relatively new. The retrospective, non-controlled format is a limitation, which is rightfully mentioned by the authors.

Some points need to be considered and adjusted before this paper is suitable for publication in PlosOne.

General comments:

1. The main risk factor associated with HEV seroprevalence is (older) age. This has been presented in many studies and is understandable considering the fact that if a person seroconverts, this is for life. So there is an accumulation of HEV seropositives in older age cohorts, also in these IBD patients. The other described risk factor in this study is the duration of anti-interleukin 12/23 treatment. This is described in the last part of Table 2, with a median duration of 0 days for both anti-HEV IgG negatives and anti-HEV IgG positives, but the ranges of duration differ. This factor is only found in the CD patients but not for the UC patients (Table 4). Actually, the multivariate analysis in Table 3 shows a 95%CI starting with 1.000. So this finding is at best a very weak relation. This finding is elaborated on in a paragraph in the Discussion. I think it should be added there that the association is very weak.

2. As mentioned by the author, another limitation is that no data are present on food intake. Especially since this is an IBD study those data would have been interesting. Both in the introduction and discussion more references can be incorporated on the relation between food intake-following a specific dietary (vegetarian) lifestyle- and HEV serostatus. See for example references such as:

- Slot et al, Meat consumption is a major risk factor for hepatitis E virus infection. PLoS One. 2017;12.

-Alberts et al, Hepatitis E virus seroprevalence and determinants in various study populations in the Netherlands, PlosOne 2018

-Wong et al, The study of seroprevalence of hepatitis E virus and an investigation into the lifestyle behaviours of the aborigines in Malaysia. Zoonoses Public health, 2020.

3. Another factor which might be of influence on the detected seroprevalence is the HEV IgG test used. In this study it is from the company Mikrogen. Please look at studies that discuss the choice of the HEV serology tests (for example Pas et al, J. Clin Virol 2013; Li et al, Liver Int 2020) and add a sentence on this in the Discussion.

Specific comments

Abstract

-In Methods, the second sentence starts with a number which is not very comprehensive. Consider rephrasing this sentence: Of patients visiting the IBD outpatient clinic….

-Consider also mentioning in the background what the HEV seroprevalence (or the range) is for a ‘general population’ in Germany in the abstract.

Introduction

In the first paragraph especially blood donor populations are mentioned. To compare German data to those of other countries also Dutch data may be relevant. Please consider incorporating some of the following studies:

-Hogema et al. Transfusion 2016; Sadik et al. BMC-ID 2016; Mooij et al. BMC-ID 2018

Results

-In the text ‘mean’ is mentioned, for example in the second paragraph: ‘mean age’ and ‘mean disease duration’, whereas this is ‘median’ in Table 1. Median, as is also mentioned in Methods, is to be preferred. Please make adjustments everywhere in the text where applicable.

- Second paragraph, second line: please change into ‘biological therapy and 65.5% had'.

- Tables 2 and 4: Please adjust ‘Age at diagnosis’

Discussion

Fourth paragraph, please consider changing the text into: ‘a greater susceptibility to HEV and higher infection rate’.

6. PLOS authors have the option to publish the peer review history of their article (what does this mean?). If published, this will include your full peer review and any attached files.

Reviewer #1: No

Reviewer #2: **Yes: **Sylvia Bruisten

---

## [Author Response · Author response to Decision Letter 0]

11 Sep 2020

 The changes were made. Now the manuscript meets PLOS ONE’s style requirements, including those for file naming.

2. Please provide the full name of the ethics committee which approved this study in the methods section of your manuscript.

 The full name of the ethics committee was provided in Methods, page 5.

3.Thank you for stating the following in the Competing Interests section:

[Annika Gauss received travel fees from Abbvie and Janssen, speaker’s fees from Janssen, MSD, Tillotts, and Takeda, and consultation fees from Janssen and AMGEN; Peter Hoffmann received travel fees from Abbvie and Janssen and speaker’s fees from Janssen; which might lead to a conflict of interest. All other authors have no conflicts of interest to declare.].

 The confirmation was done, p1.

 The updated Competing Interests statement is now included in the cover letter.

Reviewer #1: This study investigates the anti HEV seroprevalence in patients with infallamtory bowel disease. The anti HEV seroprevalence in patients with M. Crohn was lower in comparison to patients with Colitis ulcerosa. I calculated the p-value for this finding (Chi Square) and this gave 0.05, borderline significance....

The result of the statistical comparison has now been included in the discussion section on page 14. Due to the fact that the difference between HEV seroprevalences in patients with Crohn’s disease and those with ulcerative colitis was only borderline , we decided to not overemphasize it. Thus, the statistical comparison was not included in the abstract. 

A comparison to previously published cohorts of the German general population or healthy blood donors is difficult. The Mikrogen tests have been strongly improved in the last 5 years and currently these assays have a high sensitivity and specifity..... but 10 years ago these serological tests had some problems..... thus you cannot compare all published data easily....

Furthermore there are geographical variations within Germany....

if you want to compare your data with healthy people, you need your own cohort and not previously published data from various areas, using various assays....

Thank you for this important comment. Unfortunately we currently do not have our own healthy cohort to compare as a control group. 

The seroprevalence was associated with anti-interleukin-treatment..... Patients getting this treatment are usually suffering from a more severe course, thus it is more likely they get more frequently blood products or other medical interventions harboring the risk of HEV transmission.

This important consideration has now been included in the discussion section on page 14.

General testing of blood donations for HEV has been started in Germany in January 2020.

Thank you very much for this comment. The fact has been corrected in the introduction, page 4.

If you want to detect chronic HEV infection you need to test by PCR.... I suggest to do this to improve the project....

Thank you for this advise. We tested HEV PCR to detect chronic HEV infection (page 6)

Reviewer #2: This is a well written manuscript which describes a straightforward study on the possible relation between HEV seroprevalence and inflammatory bowel disease (IBD) in two groups of patients, those with Crohn’s disease (CD) and those with ulcerative colitis (UC).

The data are relatively new. The retrospective, non-controlled format is a limitation, which is rightfully mentioned by the authors.

Some points need to be considered and adjusted before this paper is suitable for publication in PlosOne.

General comments:

1. The main risk factor associated with HEV seroprevalence is (older) age. This has been presented in many studies and is understandable considering the fact that if a person seroconverts, this is for life. So there is an accumulation of HEV seropositives in older age cohorts, also in these IBD patients. The other described risk factor in this study is the duration of anti-interleukin 12/23 treatment. This is described in the last part of Table 2, with a median duration of 0 days for both anti-HEV IgG negatives and anti-HEV IgG positives, but the ranges of duration differ. This factor is only found in the CD patients but not for the UC patients (Table 4). Actually, the multivariate analysis in Table 3 shows a 95%CI starting with 1.000. So this finding is at best a very weak relation. This finding is elaborated on in a paragraph in the Discussion. I think it should be added there that the association is very weak.

The weakness of the association has now been emphasized in the discussion, page 15.

2. As mentioned by the author, another limitation is that no data are present on food intake. Especially since this is an IBD study those data would have been interesting. Both in the introduction and discussion more references can be incorporated on the relation between food intake-following a specific dietary (vegetarian) lifestyle- and HEV serostatus. See for example references such as:

- Slot et al, Meat consumption is a major risk factor for hepatitis E virus infection. PLoS One. 2017;12.

-Alberts et al, Hepatitis E virus seroprevalence and determinants in various study populations in the Netherlands, PlosOne 2018

-Wong et al, The study of seroprevalence of hepatitis E virus and an investigation into the lifestyle behaviours of the aborigines in Malaysia. Zoonoses Public health, 2020.

The above-mentioned references have been inserted in the introduction or discussion.

3. Another factor which might be of influence on the detected seroprevalence is the HEV IgG test used. In this study it is from the company Mikrogen. Please look at studies that discuss the choice of the HEV serology tests (for example Pas et al, J. Clin Virol 2013; Li et al, Liver Int 2020) and add a sentence on this in the Discussion.

This problem is now addressed in the discussion section, page 14.

Specific comments

Abstract

-In Methods, the second sentence starts with a number which is not very comprehensive. Consider rephrasing this sentence: Of patients visiting the IBD outpatient clinic….

-Consider also mentioning in the background what the HEV seroprevalence (or the range) is for a ‘general population’ in Germany in the abstract.

The changes have been made, as suggested, page 2.

Introduction

In the first paragraph especially blood donor populations are mentioned. To compare German data to those of other countries also Dutch data may be relevant. Please consider incorporating some of the following studies:

-Hogema et al. Transfusion 2016; Sadik et al. BMC-ID 2016; Mooij et al. BMC-ID 2018

Two of the suggested references have been added added in the introduction.

Results

-In the text ‘mean’ is mentioned, for example in the second paragraph: ‘mean age’ and ‘mean disease duration’, whereas this is ‘median’ in Table 1. Median, as is also mentioned in Methods, is to be preferred. Please make adjustments everywhere in the text where applicable.

- Second paragraph, second line: please change into ‘biological therapy and 65.5% had'.

- Tables 2 and 4: Please adjust ‘Age at diagnosis’

Changes were made as advised.

Discussion

Fourth paragraph, please consider changing the text into: ‘a greater susceptibility to HEV and higher infection rate’.

Changes were made as advised.

---

## [Editor Report · Decision Letter 1]

15 Sep 2020

Hepatitis E seroprevalence in a German cohort of patients with Inflammatory Bowel Diseases

PONE-D-20-21089R1

Dear Dr. Hoffmann,

We’re pleased to inform you that your manuscript has been judged scientifically suitable for publication and will be formally accepted for publication once it meets all outstanding technical requirements.

Kind regards,

Yury E Khudyakov, PhD

Academic Editor

PLOS ONE
---

## [Editor Report · Acceptance letter]

25 Sep 2020

PONE-D-20-21089R1 

Hepatitis E seroprevalence in a German cohort of patients with Inflammatory Bowel Diseases 

Dear Dr. Hoffmann:

I'm pleased to inform you that your manuscript has been deemed suitable for publication in PLOS ONE. Congratulations! Your manuscript is now with our production department. 

Kind regards, 

on behalf of

Dr. Yury E Khudyakov 

Academic Editor

PLOS ONE